# Macrogenomics-Based Analysis of the Effects of Intercropped Soybean Photosynthetic Characteristics and Nitrogen-Assimilating Enzyme Activities on Yield at Different Nitrogen Levels

**DOI:** 10.3390/microorganisms12061220

**Published:** 2024-06-18

**Authors:** Liqiang Zhang, Yudi Feng, Zehang Zhao, Bate Baoyin, Zhengguo Cui, Hongyu Wang, Qiuzhu Li, Jinhu Cui

**Affiliations:** 1College of Plant Science, Jilin University, Changchun 130012, China; lqzhang23@mails.jlu.edu.cn (L.Z.); fengyudi1@163.com (Y.F.); 18846915612@163.com (Z.Z.); bybt23@mails.jlu.edu.cn (B.B.); hong_yu@jlu.edu.cn (H.W.); 2Soybean Research Institute, Jilin Academy of Agricultural Sciences, Changchun 130033, China; 17643346860@163.com

**Keywords:** soybean intercropping, photosynthetic characteristics, nitrogen-assimilating enzymes, bacterial community, yield

## Abstract

Currently, China’s soybean self-sufficiency rate is only 15%, highlighting the soybean crisis and the supply chain risks that pose a major threat to China’s food security. Thus, it has become imperative to step up efforts to boost soybean production capacity while promoting the green and sustainable development of regional farmland ecosystems. In this context, the present study comprehensively investigated the effects of intercropping and nitrogen application rate on soybean yield, as well as the changes in gradients generated by different levels of nitrogen application. Based on six consecutive years of maize–soybean intercropping planting patterns, the inter-root soils of soybeans were collected at the flowering stage and evaluated for soil nitrogen content, nitrogen-assimilating enzyme activities, and microbial community composition of soybean, which were correlated with yield, to clarify the main pathways and modes of intercropping effects. The N_2_ level (80 kg·ha^−1^) was favourable for higher yield. In comparison to monocropping, the intercropping reduced yield by 9.65–13.01%, photosynthetic characteristics by 1.33–7.31%, and plant nitrogen-assimilating enzyme activities by 8.08–32.01% at the same level of N application. Likewise, soil urease and catalase activities were reduced by 9.22 and 1.80%, while soil nitrogen content declined by an average of 6.38%. *Gemmatimonas* and *Bradyrhizobium* enrichment significantly increased soil nitrogen content, photosynthetic characteristics, and soybean yield, while it was reduced by *Candidatus_Udaeobacter* and *Candidatus_Solibacte* enrichment. The results of this study provide a theoretical basis for further optimising maize–soybean intercropping, which is crucial for enhancing the agricultural production structure and improving the overall soybean production capacity.

## 1. Introduction

Intercropping has a long history in China, and as agricultural productivity has increased, intercropping methods and technologies have also advanced significantly [1]. However, China’s annual soybean imports have risen to 100 million tonnes since 2010. With only 16 million tonnes of domestic production, the imports account for 85% of the total [2]. In recent years, the international volatility and the impact of the new crown pneumonia epidemic have posed a huge threat to China’s food security, including soybeans, making it crucial to increase its domestic production via multiple channels. Thus, China’s soybean industry needs to be revitalised as a priority. Under the current conditions of arable land use, the expansion of the area under soybean production is limited. Nevertheless, via the development of cropping systems adapted to local conditions and soybean intercropping, the area under soybean can be indirectly increased, thus elevating China’s total soybean production [3].

Soybean provides most of the nitrogen needed for maize growth and development via nitrogen fixation, and the amount of fixed nitrogen transported to maize ranges from 25 to 155 kg·ha^−1^ [4]. In the intercropping system, maize benefited from the nitrogen uptake, which was 57.53% higher than that under monoculture, resulting in a 47.02% increase in the biological yield. However, the yield decreased by 14.56% for soybean, as being the nitrogen contributor, its uptake under intercropping was 1.21% lower than that under monoculture [5]. One of the main reasons is that the massive demand for nitrogen by maize was satisfied by the nitrogen fixed by soybeans. Consequently, the N content of the soil was reduced, which alleviated the inhibitory effect of the high N content on the activity of N-fixing enzymes. This improved the N fixation efficiency of soybean rhizomes and mitigated the disadvantageous position of soybeans in the intercropping system [6].

Increased nitrogen fertiliser application can enhance yields, but excessive usage can impair the soybean’s nitrogen fixation capacity, termed “nitrogen repression”, resulting in significant nitrogen losses [7]. Maize–soybean intercropping can positively favour the nitrogen balance of the intercropping system, and the contribution of nitrogen fixation to the whole system remains significant under normal growth of soybeans [8]. In intercropping with maize, which has an advantage in nitrogen competition, a large amount of mineral nitrogen is taken up by it. This maintains the mineral nitrogen in the soil at a low level, which reduces the inhibitory effect of mineral nitrogen on nitrogen fixation by soybean, referred to as the “slowing effect” of “nitrogen repression”. The “mitigation effect” is constituted by the following mechanisms. Maize stimulates nodulation and nitrogen fixation in soybeans, which may be due to the competitive use of nitrate and ammonium nitrogen by the maize crop in the soybean inter-root. Soybean relies more on nodular nitrogen fixation to meet its nitrogen needs when soil nitrogen levels are low. Furthermore, soybean mineral uptake is enhanced in intercropping systems, thereby increasing the amount of nitrogen fixation [9].

Plants under intercropping patterns exhibit higher yields and nutrient uptake, an advantage that further affects the activities of nitrogen-assimilating enzymes [10]. The level of nitrate reductase and nitrite reductase activities affect NO_3_^−^ accumulation. The glutamine synthetase/glutamate synthase cycle is the critical pathway for ammonia assimilation in higher plants [11]. Soil enzymes play a vital role in maintaining soil ecological balance and are indicators of soil ecosystem function [12]. For example, intercropping maize with soybean and groundnut significantly increased urease and catalase, respectively [13,14]. Urease promotes the hydrolysis of urea to ammonium nitrogen and CO_2_, providing plants with directly absorbable nitrogen and promoting their growth [15].

Previous studies have shown that maize–soybean intercropping significantly increased urease activity in soil layers with different profiles from 0 to 40 cm. The urease activity was significantly correlated with organic carbon and total nitrogen [16]. Catalase, an oxidoreductase widely distributed in soils mitigates H_2_O_2_-induced oxidative damage to plant roots [17]. As compared to maize monoculture, the maize–soybean intercropping significantly increased soil peroxidase activity [18]. However, most studies on soil enzymatic activities have focused on different cropping practices and soil fertility [19].

Building upon the aforementioned literature, it was postulated that the alterations in cropping patterns caused by nitrogen utilisation in maize–soybean intercropping might be closely linked to key species within the photosynthetic characters and soil’s microbial community. Currently, the research on maize–soybean intercropping is mainly focused on optimising cropping technology and improving maize yield [20,21], while the specific traits of soybeans are less explored. Specifically, only above-ground traits such as agronomic traits, quality traits, and yield traits were analysed under intercropping patterns [22]. Furthermore, the effect of changes in the soil microenvironment on soybean yield has not received much attention. The present study presents a comprehensive analysis of above-ground and below-ground traits under a maize–soybean intercropping system for six consecutive years. The objectives of this study comprised studying the effect of the maize–soybean intercropping system on (1) the photosynthetic characteristics of soybean; (2) the nitrogen content of soybean soil, and the activity of key enzymes of nitrogen transformation; (3) the changes in soil microorganisms in the inter-root of soybean; (4) the critical factors and mechanisms affecting the soybean yield. This study is of great significance in guiding the yield potential of soybeans under intercropping mode.

## 2. Materials and Methods

### 2.1. Study Area

The experiment was conducted at the Agricultural Experimental Base of Jilin University, Changchun City, Jilin Province (E125°14.23′, N43°56.60′). The environmental parameters included black soil, temperate continental semi-moist monsoon climate, altitude of 245 m above sea level, and an average annual precipitation of 600–700 mm. The average annual temperature was 4.6 °C, with extremes of 40 °C and −36.5 °C. The annual frost-free period was 140–150 days, while the annual freezing period was 150–160 days. The soil type in the field is Phaeozems (FAO-WRB classification system, 2014). The soil attributes comprised of initial pH of 5.33, total organic carbon of 1.37%, total nitrogen of 1.41 g·kg^−1^, total phosphorus of 0.48 g·kg^−1^, and total potassium of 21.42 g·kg^−1^.

### 2.2. Experimental Design

The experiment used a two-factor split-zone design, with the level of nitrogen supply as the main zone and the planting pattern as the split zone. The tested soybean variety was Changnong 16, while the maize was Xianyu 335. Both varieties were provided by the Jilin Academy of Agricultural Sciences. The seeds were sown on 26 April 2023, with soybean monoculture (MS) as the control group and maize–soybean intercropping (IS) as the experimental group. Each plot had an area of 62.4 m^2^, with 12 rows of soybeans in the monoculture plot. The maize–soybean intercropping plot consisted of three strips. Each strip was planted with two rows of maize and two rows of soybeans. The row spacing was 65 cm, with a soybean density of 180,000 plants/ha and a maize density of 90,000 plants/ha (Figure 1).

Fertiliser application in intercropping was the same as in monocropping, with soybean receiving a basal application of 0, 40, 80, and 120 kg N ha^−1^, and maize receiving 0, 180, 240 and 300 kg N ha^−1^, with basal applications of 0, 40, 80 and 120 kg N ha^−1^. The rest of the nitrogen fertiliser was applied in two separate applications at the corn-pulling stage and the big trumpet stage (4:6), respectively. The phosphate fertiliser for all treatments was heavy calcium superphosphate (P_2_O_5_ 46%), applied at 120 kg P_2_O_5_ ha^−1^. Potash in the form of potassium sulphate (K_2_O 50%) was applied at 100 kg K_2_O ha^−1^. Both phosphate and potash were applied as basal dressings.

### 2.3. Sample Collection

Topsoil (0–20 cm, adjacent to soybean on the ridge) and rhizosphere soil samples were collected on 21 July 2023 (soybean flowering period). The surface soil samples were subjected to chemical analysis, and soil urease and catalase activities were determined after air drying. The rhizosphere soil samples were stored in the refrigerator at −80 °C for microbial diversity determination. Before harvesting, two representative rows were chosen consistently in each plot of mature soybeans. Following threshing, the grain weight was measured, which was converted into hectare yield according to the harvested area.

### 2.4. Measurements and Methods

#### 2.4.1. Photosynthetic Characteristics

During the flowering period, the SPAD value of soybean leaves was measured using a hand-held chlorophyll meter (LD-YD, 0.0–99.99 SPAD, ShiYa, Shijiazhuang, China). Net photosynthetic rate (Pn) and transpiration rate (IR) were measured using a photosynthesiser (Li-6800, LI-COR, Lincoln, Dearborn, MI, USA) at 9:00–11:00 on a sunny day during the same period. The middle part of the leaf blade of inverted trifoliate leaves of soybean was selected to avoid the leaf veins, and three points along the left and right of the veins were measured and counted as the average. The Intelligent Leaf Area Measurement System (YMJ-CHA3, TOP CLOUD-AGRI, Hangzhou, China) was used to measure the leaf area of three soybean plants.

#### 2.4.2. Determination of Nitrogen-Assimilating Enzymes

The inverted trifoliate leaves were collected at the flowering stage and placed in a liquid nitrogen tank after removal of veins. The respective standard kits were used to extract plant glutamate synthase (CAS: NONE18162, ELISA kit, Shanghai, China), nitrite reductase (CAS: NONE18350, ELISA kit, Shanghai, China), nitrate reductase (CAS: NONE18344, ELISA kit, Shanghai, China), soil urease (CAS: NONE16585, ELISA kit, Shanghai, China) and soil catalase (CAS: NONE16574, ELISA kit, Shanghai, China). The activity titre was determined using an enzyme label (Feyond-A300, 0–4 OD, ALLSHENG, Shanghai, China) at respective wavelengths of 340 nm, 540 nm, 340 nm, 630 nm, and 240 nm.

#### 2.4.3. Soil Chemical Properties

To determine soil pH and EC, the water-soil ratio of 5:1 was shaken at 180 r/min for 5 min and left for 30 min. Thereafter, pH and EC were assessed with a pH meter (pH-100A, 100–2000 rpm, LICHEN, Shanghai, China) and a conductivity meter (DDSJ-11A-307, YUEPING, Shanghai, China), respectively. Soil ammonium nitrogen (NH_4_^+^-N) and nitrate nitrogen (NO_3_^−^-N) fractions were extracted with sodium bicarbonate, shaken at 180 r/min for 2 h, allowed to stand for 30 min, and filtered through a 0.45 µm PES membrane. Soil total nitrogen (TN) was digested by the Kjeldahl method and filtered through a 0.45 µm PES membrane. The above samples were analysed using a continuous flow analyser (AA3, AutoAnalyzer 3, Technicon, Windows/NT, SEAL, Harbin, China) [23]. Soil total organic carbon (TOC) fractions were determined by wrapping 10 mg of soil samples in aluminium paper and subjecting them to an elemental analyser (Vario TOC cube, NDIR, 60,000 ppm, 2 ppb, Elementar, Shanghai, China) [24].

#### 2.4.4. Diversity of Soil Bacterial Communities

DNA was extracted from soil samples using a DNA kit (MN NudeoSpin 96 Soi), and the concentration was determined using NanoDrop 2000. The PCR reaction conditions were as follows: pre-denaturation at 94 °C for 5 min, 30 cycles at 94 °C for 30 s, 50 °C for 30 s and 72 °C for 60 s. The reaction was followed by a stable extension of 72 °C for 7 min and final storage at 4 °C. Extracted genomic DNA was visualised by 1% agarose gel electrophoresis. For bacterial 16S, primers 338F (5′-ACTCCTACGGGAGGCAGCAG-3′) and 806R (5′-GGACTACHVGGGTWTCTAAT-3′) were used to amplify the V3-V4 region, and the products were purified, quantified, and standardised. The library construction comprised: (1) ligation of the “Y” junction; (2) removal of junction self-associated fragments by magnetic bead screening; (3) enrichment of the library template by PCR amplification; (4) denaturation by sodium hydroxide to produce single-stranded DNA fragments. Hardware and software information about sequencing is shown in Table 1.

Sequencing constituted of (1) one end of the DNA fragment was complementary to the primer base and fixed on the chip; (2) the other end was randomly complementary to another primer in the vicinity, which was also fixed, forming a “bridge”; (3) PCR amplification produced DNA clusters; (4) the DNA amplicon was linearised into a single strand; (5) the modified DNA polymerase and dNTP were added with four fluorescent labels, synthesising only one base per cycle; (6) the surface of the reaction plate was scanned with a laser to read the nucleotide species polymerised in the first reaction of each template sequence; (7) the “fluorescent group” and the “termination group” were chemically cleaved to restore 3′-end attachment, and continued to polymerise the second nucleotide; (8) the fluorescence signals collected in each round were counted to obtain the sequence of the template DNA fragment [25]. All the deposition of sequences and expression data has been uploaded to the NCBI (National Center for Biotechnology Information) (https://www.ncbi.nlm.nih.gov/) URL (accessed on 13 June 2024). Accession numbers: PRJNA1123432.

### 2.5. Statistical Analysis

Statistical analyses were performed using SPSS 22.0 software. Two-way ANOVA was used to compare the effects of intercropping on photosynthetic characteristics, soil parameters, and yield. After evaluating the significant differences between the sample means via one-way ANOVA, we used Duncan’s test, a post hoc test, to determine which specific group means were critically different from each other. To evaluate the relationships between the relative abundance of dominant genera, soil chemical properties and yield, the Pearson correlation test was used. Beta diversity analysis was performed based on the coefficient of variation of the Aitchison distance. The PCA (principal component analysis) was conducted to compare the degree of similarity that existed between different samples in terms of species community diversity. Significantly different species among different groups were analysed and compared by LEfSe. Microbial ecological networks and topological indices were visually analysed using Gephi software (version 0.9.6).

The following topological indices were used to describe the nodes and connecting lines in the network. (1) The number of connecting lines of a node, which is the number of all connecting lines connected to each node; (2) the median centrality of a node, which is the node located on the shortest path between two nodes, calculated as in the formula (a); (3) the topological coefficient of a node, which embodies the proximity of the nodes and is expressed by the formula (b); (4) the connecting line weights, which reflect the number of connections between a particular OTU (operational taxonomic unit) node and the sample node; (5) the connecting line centrality, a parameter that shows the importance of each connecting line in the information transfer process of the whole network [26]. Structural equation modelling (SEM) of the effects of intercropping and nitrogen application rates on yield pathways was constructed using R version 4.3.1 (https://www.r-project.org/) URL (accessed on 1 June 2024).
(1)$$Cb\left(n\right)=\sum_{s\neq n\neq t}\left(\frac{\sigma_{st}\left(n\right)}{\sigma_{st}}\right)$$
where *n* is the destination node, *s* and *t* are nodes in the network other than *n*, $\sigma_{st}$ represents the number of shortest paths from node *s* to node *t*, and $\sigma_{st}(n)$ denotes the number of shortest paths from node *s* to node *t* that must pass through node *n*.
(2)$$T_{n}=\frac{avg\left(J_{\left(n,m\right)}\right)}{k_{n}}$$
where $J_{\left(n,m\right)}$ is the number of all nodes adjacent to both nodes *n* and *m*, and the value of $J_{\left(n,m\right)}$ is increased by 1 if *n* is directly adjacent to m. $k_{n}$ is the number of all neighbours of the node.

## 3. Results

### 3.1. Photosynthetic Traits and Yield

To study the effects of intercropping (IS) and different levels of N application on soybean photosynthesis, the leaf area (LA), leaf SPAD values, net photosynthetic rate (Pn), and transpiration rate (IR) of soybean were measured at the flowering stage (Table 2). The analysis showed that the Pn, IR, and SPAD values of soybeans were lower than those of monoculture soybeans (MS). At the N0 level, the parameters under IS were reduced by 7.31, 6.74 and 1.33%, respectively, compared to MS. The Pn rose with increasing N application, while IR and SPAD reached their maximum values at MSN1 and ISN2. Any further increase in N application led to a decrease in IR and SPAD. However, at the same level of N application, LA was higher in both IS models than in MS, with an increase of 61.07% in IS compared to MS at the N0 level. With increasing N application, both IS and MS reached the maximum value at the N2 level.

Comparison of yield variations between IS and MS at different N application levels showed that at the same N application level, soybean yield in IS mode was lower than that in MS. Nevertheless, with increasing N application, soybean yield in MS mode continued to increase, but soybean yield in IS mode was highest at N2 level. At the N0 level, soybean yield in IS mode was 13.01% lower as compared to MS mode. However, this difference reduced to 9.65% with increasing N application to N2 level. A two-factor ANOVA showed that N application rate and N*C (IS of N application rate and cropping pattern) primarily affected SPAD and yield, while the cropping pattern had no effect on soybean SPAD values. Overall, IS reduced the photosynthetic characteristics of soybeans and consequently lowered yield, but these differences were gradually reduced as the increased N application raised the N2 levels.

### 3.2. Soil Chemistry and Nitrogen Content

As seen in Figure 2a, the soil pH of the MS and IS treatments decreased with increasing N application. For both, it was lowest in the N3 treatment, which exhibited a reduction of 2.73–2.75% compared to the N0 treatment. However, there was no difference in soil pH under MS and IS at the same N level. Soil EC in each treatment under MS tended to rise and then decline with increasing N application rate (Figure 2b). There was no difference between N3 and N0 levels. Conversely, the changes in the soil EC under each IS treatment were exactly the opposite and exhibited a pattern of decrease and then increase with the rising nitrogen application levels. The highest EC was observed at the N3 level. At the levels of N1 and N2, the soil EC decreased instead and was even lower than the N0 treatment. For instance, N1 reduced the soil EC by 11.93% compared to the N0 level. Under MS and IS, soil TOC content reached its maximum with an increasing nitrogen application rate up to N2, but any further increase lowered the soil TOC content (Figure 2c). Among them, the soil TOC content was enhanced by 7.36% and 12.30% under N2 as compared to the N0 treatment. However, the soil TOC content was higher under MS than IS at the same N level, which was most significant under N0, being 7.38% higher. Overall, excessive N application accelerated soil acidification, while adequate N application increased soil TOC content, but IS had a reducing effect on both.

Figure 2d–f depicts the effect of different cropping patterns and nitrogen application levels on soil N content. The findings showed that soil NH_4_^+^-N and NO_3_^−^-N contents were lower under IS than MS. With the increase in nitrogen application, under MS, the contents of both gradually increased, reaching a maximum at the N3 level, exhibiting a respective increase of 36.55% and 43.97%, compared to N0 level. However, under IS mode, both reached their maximum values at the N2 level and increased by 28.54% and 51.10%, respectively, compared to the N0 level. Soil TN content under MS mode reached its maximum value at the N2 level, but any further increase in N application led to a reduction in soil TN content. It is worth noting that there was no change in soil TN content with increasing N application treatments under MS. A difference was found between N0 and N3 levels under IS. At N0–N2 levels, soil TN content under IS was lower than that under MS. However, at the N3 level, soil TN content under IS was higher than in MS. Overall, soil NH_4_^+^-N and NO_3_^−^-N contents decreased under IS as compared to MS, but the gap between IS and MS could be narrowed by controlling N application at N2 level under IS mode.

### 3.3. Soil Chemistry and Nitrogen Content

To evaluate the nitrogen assimilation capacity of soybean under IS, nitrate reductase (NR), nitrite reductase (NIR), and glutamate synthase (GOGAT) activities were measured in plants (Figure 3a–c) and urease (UE) and catalase (CAT) in soil (Figure 3d,e). Overall, NR, NIR, and GOGAT activities were higher in the MS than in the IS treatment at the same level of nitrogen application. In the MS mode, the plant NR, NIR, and GOGAT activities were highest at the N2 level and increased by 36.87%, 8.96%, and 19.26%, respectively, compared to the N0 treatment. However, any further increase in nitrogen application led to a decrease in their activities. In IS mode, NIR and GOGAT activities first rose and then dropped with increasing N application. The NIR activity was highest at the N1 level, exhibiting an increase of 17.97% compared to the N0 level. Likewise, GOGAT had the highest activity at the N2 level, reflecting an increase of 52.27% compared to the N0 level.

The soil NR activity grew with increasing N application, and its rise was most prominent at the N0–N1 level, with an increase of 50.97%. The effect was less pronounced at the N1–N2 and N2–N3 levels, with an enhancement of 20.22% and 10.38%, respectively. The soil UE and CAT activities were higher in IS as compared to MS. With increasing N application, soil UE and CAT were highest at the N2 level regardless of the cropping system. Under N2 application, the soil UE activity grew considerably by 14.54−22.74%, while CAT activity increased by 1.18–2.48% as compared to the N0 level. Overall, in comparison to MS, IS reduced the nitrogen-assimilating enzyme activity of soybean plants but enhanced soil nitrogen-assimilating enzyme activity. On combining all data, the highest nitrogen-assimilating enzyme activity was observed at the N2 level under both MS and IS.

### 3.4. Dynamics of the Soil Microbial Communities

#### 3.4.1. Alpha Diversity of the Bacterial Community

After the completion of the sequencing, OTUs were screened for de-low content. Non-repetitive sequences (excluding single sequences) were clustered into OTUs based on 97% similarity, and chimaeras were removed in the clustering process to obtain representative sequences of OTUs. The final number of OTUs was 3601, resulting in at least 35,257 optimised sequences per sample. To investigate the diversity of individual soil samples (Alpha diversity), the richness (Chao1 index), diversity (Shannon index and PD_whole_tree), and sequencing depth (Goods_coverage) of the soil bacterial community were calculated separately for each treatment. As depicted in Table 3, the bacterial coverage of Goods_coverage was higher than 95% in all treatments. Chao1 and PD_whole_tree indices varied consistently across treatments but were higher in IS than MS at the same level of nitrogen application. Further analysis showed that the diversity of the bacterial community was subsequently reduced under MS.

The diversity of the bacterial community was reduced at N3 compared to N0 by 5.00% and 6.59%. However, both Shannon and PD_whole_tree indices under IS reached their maximum values at the N2 level, wherein they rose by 1.78 and 0.55%, respectively, compared to the N0 treatment. Further increase in nitrogen application at the N3 level resulted in a sharp drop, even below the N0 treatment, with decreases of 3.55% and 6.08%, respectively. Under IS, the Shannon index decreased with increasing N application, being highest at the N0 level and lowest at the N3 level, with a decrease of 2.90%. However, under MS, there was no obvious pattern of change in the Shannon index at different N application levels, and there was no difference between treatments. A two-way ANOVA showed that both nitrogen application level (N) and cropping mode (C) were significantly correlated with Alpha diversity (*p* < 0.01). Overall, IS increased the diversity and richness of soil bacterial communities, but increased nitrogen application had an inhibitory effect. Nevertheless, when nitrogen application was maintained at the N2 level under IS, the diversity of the soil bacterial community was the highest.

#### 3.4.2. Composition of Horizontal Communities of Dominant Bacterial Genera

All soil samples could be annotated with 878 bacterial genera. After selecting the top 50 genera in relative abundance and excluding the remaining 7 with an abundance of less than 1%, the dominant genera accounted for 27.95–37.28% of the total (Figure 4). At the same level of N application, the abundance of IS dominant bacterial genera was higher than that of MS, increasing by 11.64–12.96%. Nevertheless, the abundance of both MS and IS dominant bacteria was highest at the N1 level. Further comparison revealed that the relative abundance of *Sphingomonas* was highest at the N1 and N2 levels under MS and IS, respectively, accounting for 3.45% and 4.04%. The relative abundance of *Candidatus_Udaeobacter* under MS exhibited a decreasing trend with increasing nitrogen application, with 28.03% higher abundance in N0 compared to N3. However, under IS, it had the highest relative abundance at the N1 level and decreased thereafter. The N1 level abundance was greater by 48.97% and 46.82% as compared to N0 and N3 levels, respectively.

The relative abundance of *Bradyrhizobium* under MS and IS had a large difference at the N0 level, with IS being 69.03% higher than that of MS. There was no significant change in the genus under each level of nitrogen application in MS. However, under IS treatment, it exhibited an overall decreasing trend with the increase of nitrogen application, with a reduction of 31.12–38.00% in N1–N3 as compared to the N0 level. The changes in *Gemmatimonas* were the same at different levels of nitrogen application in both MS and IS and were highest at the N2 level. The relative abundance of *Gemmatimonas* at different nitrogen application levels of MS and IS was consistent, with the highest being at the N2 level. At the N0 level, the relative abundance of *Gemmatimonas* was higher in IS than in MS. When the nitrogen application was increased to the N1–N2 level, the relative abundance of *Gemmatimonas* in both MS and IS increased, but its relative abundance was greater in MS. However, at the N3 level, the relative abundance of *Gemmatimonas* decreased and was again predominant in IS as compared to MS.

The relative abundance of *Candidatus_Solbacte* was highest at N1 and N2 levels in MS and IS, respectively, with an increase of 37.42% and 48.61% compared to N0. However, the relative abundance of *Candidatus_Solbacte* rose by 24.25% at the N0 level in IS compared to MS. The relative abundance of *Saccharimonadales* showed a decreasing trend with the increase in nitrogen application under MS and IS and was 42.93% and 59.09% higher at N0 than at N3, respectively. The relative abundance of *Bryobacteria* showed a rise and drop pattern with the increase in nitrogen application under MS and IS. Its abundance was highest at N2 and N1 levels, which, respectively, rose by 2.53% and 41.88% as compared to N0. Summarily, IS increased the relative abundance of *Sphingomonas*, *Bradyrhizobium,* and *Saccharimonadales* but decreased the relative abundance of *Candidatus_Udaeobacter* and *Bryobacter*. Furthermore, regardless of cropping pattern, the N1 level was most favourable for the dominant genera of rhizosphere soil bacteria in soybean enrichment. The excessive nitrogen application (N3) disrupted the community structure and led to a decrease in the abundance of dominant bacterial genera, even below the N0 level.

#### 3.4.3. Beta Diversity of the Bacterial Community

Beta diversity analysis was performed based on Aitchison distance. PCA was chosen to compare the degree of similarity that existed in terms of species community diversity among the different samples (Figure 5). The first principal component (PC1) and the second principal component (PC2) of the treatments under different cropping patterns and nitrogen application rates accounted for 35.49% and 25.03% of the bacterial community structure (97% similarity). Different N application rates had no effect on the bacterial community structure under MS, and all treatments were clustered in the second quadrant without segregation. However, under IS low nitrogen levels (N0 and N1), segregation occurred at PC1 and PC2, indicating that the two treatments differed from the other treatments in terms of community structure. Nevertheless, with the increase in nitrogen application, the two levels of N2 and N3 under IS were not significantly different from the MS.

The two MSN0-N1 treatments separated at PC1 and PC2, respectively, indicating a significant (*p* > 0.05) difference from the other treatments in terms of community structure. Overall, the MS model had no effect on the soil bacterial community structure regardless of the level of nitrogen application. Furthermore, the difference in the soil bacterial community structure under low nitrogen levels (N0 and N1) was significant (*p* < 0.05) in the IS model. However, any further increase in nitrogen fertiliser resulted in a soil bacterial community structure similar to that of the MS model.

#### 3.4.4. LEfSe Analysis of Differentiated Bacterial Communities

As depicted in Figure 6, differential species were mainly present in the N1 and N2 treatments in the IS model, and any further increase in nitrogen fertiliser reduced their number. For instance, there was only one differential species in ISN3, i.e., *Geodermatophilaceae*. There were no differential species in the ISN0, MSN2, and MSN3 treatments and therefore not reflected in the figure. The highest number of differential species were observed in ISN1, with BIrii41 exhibiting the highest LDA value, followed by nine differential species such as *Rhizobiales_Incertae_Sedis*, *TM7,* and *Nordella*. The differential species in ISN2 treatment were mainly *Ktedonobacteria*, *Ktedonobacterales,* and *Planctomycetes*, and the rest of the species had more similar LDA values. The differential species in MSN0 and MSN1 were mainly from *Musu_ABB* and *Blastococcus*. Taken together, there was no significant change in the differential species with the increase in nitrogen application in the MS mode. Nevertheless, the IS mode had the highest number of different species at the N1-N2 levels, indicating that the rhizosphere soil environment of soybean was more suitable at this level of nitrogen application.

#### 3.4.5. Correlation Analysis of Dominant Bacterial Genera with Soil Environment and Yield

The relative abundance of each dominant genus was statistically correlated with photosynthetic characteristics, soil environment, and soybean yield (Figure 7). The relative abundance of *Saccharimonadales* exhibited a highly significant negative correlation with yield (*p* < 0.01) (Figure 7a). The enrichment of this bacterial genus significantly reduced the Pn and SPAD values of soybean (*p* < 0.05). Enrichment of *Sphingomonas*, *Gemmatimonas,* and *Bradyrhizobium* significantly elevated soybean yield (*p* < 0.05). The relative abundance of these three genera was positively correlated with most of the soybean LA, SPAD values, Pn, and transpiration rate (IR). *Candidatus_Udaeobacter* relative abundance had a highly significant negative correlation with LA (*p* < 0.01). Contrarily, the relative abundance of *Bryobacter* and *Candidatus_Solbacte* had no significant effect (*p* > 0.05) on soybean photosynthetic characteristics and yield.

Figure 7b illustrates the correlation with soil factors. The relative abundance of *Sphingomonas* demonstrated a highly significant negative correlation with soil pH (*p* < 0.01). The genus also exhibited a significant negative correlation with plant NIR but a significant positive correlation with soil UE (*p* < 0.05). *Gemmatimonas* and *Bradyrhizobium* enrichment significantly raised soil nitrogen content. The relative abundance of *Gemmatimonas* had a highly significant positive correlation with plant GOGAT, soil CAT, and UE activities. The *Bradyrhizobium* relative abundance exhibited a highly significant negative correlation with NIR activity (*p* < 0.01). The enrichment of two genera, *Candidatus_Udaeobacter* and *Candidatus_Solbacte*, could reduce soil N content, with the relative abundance of *Candidatus_Solbacte* showing a significant negative correlation with NO_3_^−^-N (*p* < 0.05).

The *Candidatus_Udaeobacter* enrichment mainly affected soil total N content and had a highly significant negative correlation with soil UE activity but a highly significant positive correlation with soil CAT activity (*p* < 0.01). *Bryobacter* enrichment significantly raised soil TOC content but showed a highly significant negative correlation with soil EC (*p* < 0.01). *Saccharimonadales* mainly affected plant NIR and NR activities. The findings revealed a highly significant positive correlation (*p* < 0.01) with NIR, a significant positive correlation with NR, and a significant negative correlation (*p* < 0.05) with soil CAT activity.

#### 3.4.6. Co-Occurrence Network Modelling of Soil Bacterial Communities

To elucidate the mechanism of synergistic interactions among genera, the co-occurrence network models of soil bacterial communities were constructed at the genus level for the top 200 abundances of bacterial genera level for different nitrogen application levels (MS and IS fitted by 97% similarity) (Figure 8a–d) and for different cropping modes (N0–N3 fitted by 97% similarity) (Figure 8e,f), respectively. The topology parameter list of the network models (Table 4) was counted to compare the interconnections among soil bacterial communities.

The analysis showed that the number of edges and positive correlation percentage of soil bacterial community decreased by 11.97–19.76% and 2.80–4.03% in N1–N3 levels compared to N0 levels. However, under IM, the number of edges rose by 72.42% and the positive correlation percentage decreased by 2.28% as compared to the MS planting pattern. It indicated that IS and nitrogen application promoted the competitive relationship between species while weakening the synergistic ability. IM also promoted the interconnection between genera and complicated the interrelationship between genera.

The average degree, average weight, average clustering coefficient, and modularity parameters were compared. The findings revealed that, except for the modularity parameter, all network topology parameters at the N1–N3 nitrogen application level were lower than the N0 level and were higher for IM as compared to MS. This indicated that the degree of connectivity between network nodes was stronger under the IS cropping mode, and the connections between nodes were more numerous and complex.

#### 3.4.7. Structural Equation Modelling-Based Analysis of Intercropping and Nitrogen 

##### Application Rates on Soybean Yield Trajectories

The results of structural equation modelling (Figure 9) revealed that IS negatively impacted photosynthetic characteristics, nitrogen-assimilating enzymatic activities, and soil nitrogen content. However, there was some boost to the composition of the rhizosphere soil bacterial community. The N fertilisation had a positive relationship with photosynthetic characteristics, N-assimilating enzyme activity, soil N and the composition of the rhizosphere soil bacterial community. Although N fertiliser application increased the yield of intercropped soybeans to a certain extent, it was not enough to offset the yield-reducing effect of IS. The primary reason for yield loss during IS was the reduction in photosynthesis and soil nitrogen content under intercropping. Although soybean yield was reduced under the intercropping mode, it effectively promoted bacterial community composition and function. This is of great value in maintaining soil health and providing a favourable soil environment for crop growth.

## 4. Discussion

### 4.1. Effects of Intercropping and Different Nitrogen Rates on Photosynthetic Characteristics and Yield of Soybean

Intercropping systems may leverage the complementing benefits of two crops, resulting in the inhibition of the growth or development of at least one crop, and taller crops generally restrict the growth of shorter crops [27]. This study’s findings showed that Pn, IR, SPAD values, and soybean yield were lower in the IS than in the MS treatment. The primary reason can be attributed to the constraint of providing an adequate light environment. Intercropping reduces the light energy resources available to soybeans, which indirectly reduces the photosynthetic rate of soybeans under natural light. This, in turn, affects the growth and development of soybeans, resulting in lower yield [28]. Previous studies have reported a significant reduction in the net photosynthetic rate and transpiration rate of soybean leaves by intercropping, and the reduction in soybean biomass was determined by the leaf area index [29]. Contrarily, in this study, the leaf area of soybean under IS was higher than that of MS. This could be due to the stronger ventilation conditions and space occupied by a single plant under intercropping conditions than monocropping, which led to an increase in the leaf area of soybean [30].

Previous research has found that leaves chlorophyll has a positive effect on the photosynthetic rate. The increase in chlorophyll content significantly improved the ability of chloroplasts to convert light energy, resulting in enhanced photosynthetic rate and yield [31]. The results of this study are in agreement with these findings. Via varying levels of N application, it was found that the best photosynthetic characteristics and yield were achieved when the N application of maize–soybean intercrop was increased to the N2 level. Either too low or too high N application level adversely affected the photosynthetic characteristics and yield of soybeans. This is because higher levels of nitrogen application in C3 plants can alter leaf nitrogen and photosynthetic protein content.

At appropriate levels of nitrogen application, approximately more than half of the nitrogen in leaves is allocated to the photosynthetic system, enhancing the photosynthetic characteristics and yield of the plant [32]. As the level of nitrogen application increases, so does the chloroplast volume, raising photosynthetic capacity. However, the photosynthetic nitrogen use efficiency decreases significantly, owing to the corresponding decrease in Rubisco enzyme activity after the increase in chloroplast volume. Rubisco activity is closely related to the carboxylation rate, and the excessive application of nitrogen has an inhibitory effect on the rate of carboxylation. This limits the increase in photosynthetic rate and ultimately reduces the photosynthetic characteristics, adversely affecting soybean yield and dry matter accumulation [33].

### 4.2. Effects of Intercropping and Different Levels of Nitrogen Application on Soybean Soil N Content and N-Assimilating Enzyme Activity

Soil nitrogen is the main source of nitrogen uptake and utilisation by plants and thus, a key element in attaining high-yielding crops [34]. In legume-non-legume intercropping, legumes can efficiently fix nitrogen that is utilised by the same crop via various transfer pathways or residues utilised by subsequent non-legume crops. This pattern plays a very important role in conventional agriculture [35]. Soil NH_4_^+^-N and NO_3_^−^-N are nitrogen forms that can be used directly by plants, and thus, enhancing their levels facilitates the uptake and use of nitrogen by plants [36]. In this study, intercropping reduced soil TN, NH_4_^+^-N, and NO_3_^−^-N contents as compared to soybean monoculture.

Maize has a high fertiliser demand and strong fertiliser uptake capacity. Owing to the differences in morphological structure and characteristics of maize and soybean, the nitrogen fixation capacity of soybean was inhibited to a certain extent in the intercropping system. Although soybeans could fix a certain amount of nitrogen, it could not satisfy the growth of the plant. Thus, the plant extracted nitrogen from the soil, which in turn resulted in lower nitrogen content in IS than in MS [37]. Application of varying nitrogen rates revealed that the highest soil nitrogen content in maize–soybean intercrop was found at the N2 level, and too low or too high nitrogen application rates had an adverse impact. Nitrogen fertiliser application and soil nitrate content are closely related to leaching. Nitrate accumulation is generally not significant with normal and appropriate levels of fertiliser application. However, excessive and inappropriate fertiliser application leads to significant accumulation and leaching of nitrate in the soil [38].

Higher plants are mainly dependent on nitrate and ammonium for nitrogen inputs. After nitrate uptake by plants, it must be converted back to ammonium before it can be further assimilated into amino acids. NR, NIR, and GOGAT enzymes are mainly involved in the biocatalytic process of ammonia assimilation in the plants [39]. In the present study, the plant NR, NIR and GOGAT activities were lower under IS than under MS, and all three tended to increase with increasing N application. Nitrogen content in soybeans affects chlorophyll content, photosynthetic electron transfer, and the formation of nitrogen-assimilating enzymes in leaves, which in turn impacts the distribution of nitrogen in soybean plants and, consequently, photosynthesis. However, the weakening of photosynthesis in soybean intercropping inevitably reduces the nitrogen demand for soybeans, leading to a decrease in nitrogen-assimilating enzyme activity [40]. Additionally, the application of nitrogen fertiliser can increase leaf nitrogen content, improve the photosynthetic rate and promote the formation of nitrogen-assimilating enzymes [41].

Soil UE and CAT are crucial for soil nitrogen accumulation and plant growth [42]. In this study, UE and CAT were higher in the root zone of soybean under IS. This could be due to the interspecific competition in the maize–soybean intercropping system, and IS had a greater competitive intensity for soil UE and CAT at the flowering stage. Furthermore, due to the high nutrient demand of maize for nitrogen, the root system of intercropped maize competed with soybean for nitrogen uptake, reducing the nitrogen content in the root zone of soybean. This, in turn, stimulated nitrogen fixation by soybean rhizobia, which in turn enhanced soil UE and CAT activities [18].

### 4.3. Effects of Intercropping and Different Levels of Nitrogen Application on the Soil Bacterial Community in Soybean Inter-Roots

Plants, via their root secretions, can actively induce the growth of some specific soil microbial communities to provide a favourable soil environment for plant growth [43]. In this study, the diversity of rhizosphere soil bacterial communities and the relative abundance of its dominant genera was higher in IS as compared to MS. However, with excessive N application, this growth trend was disrupted and there was a decrease in endemic differential species. This is because, following the excessive application of N fertiliser, plants typically use only 30–50% of N fertiliser. The remaining is converted into nitrate in the soil by nitrification, which causes soil acidification. Further, the neutralisation reaction with inorganic carbon (carbonate) in the soil, leads to a loss of inorganic carbon and destroys the soil. This inevitably impacts the diversity of the soil bacterial community. It has been shown that increasing plant diversity supports a more varied soil microbial community, which significantly improves soil fertility and reduces the number of pathogenic microorganisms [44]. Previous research has demonstrated that increasing soil bacterial diversity enhances soil nitrogen nutrient availability, which promotes plant growth as well as nitrogen use [45]. Additionally, studies have validated that the number of copies of genes involved in ecological processes related to soil nitrogen metabolism was significantly higher in healthy than in diseased soils [46].

Maize–soybean intercropping under subsoil intercropping conditions promoted the soil bacterial community diversity, the level of soil N and other nutrients supply. This helped in maintaining the stability of the soil microhabitat system while inhibiting pests and diseases, thus promoting intercrop growth and increasing yields. The relative abundance of *Sphingomonas*, *Bradyrhizobium,* and *Saccharimonadales* was higher in IS than in MS, but the relative abundance of *Candidatus_Udaeobacter* and *Bryobacter* was lower than in MS. It was found that under the intercropping mode, root secretion became the main group of Ascomycetes, of which *Sphingomonas* was the main group. This phenomenon was conducive to the ability of soybean roots to fully utilise inorganic nutrients in the soil [37]. Under intercropping conditions, the bacterial community results varied dramatically due to the significant changes in the composition of secretions in the inter-root zone.

### 4.4. Relationships between Photosynthetic Properties, Soil Nitrogen, Nitrogen-Assimilating Enzymes, Microbes, and Yield

In recent years, research efforts to study the effects of agrotechnological practices on soil microbial communities have increased exponentially [47]. However, elucidating the correlations between photosynthetic properties, soil nitrogen, nitrogen-assimilating enzymes, microbes, yield, and the pathways of influence under intercropping is still challenging. Soil microbial community structure and functional diversity are closely related to soil fertility release and soil quality improvement, which in turn affects above-ground growth and development [48].

Plant–microbe interactions between roots are closely related to the growth and yield formation of intercrops. In this study, the correlation analysis showed that the enrichment of *Gemmatimonas* and *Bradyrhizobium* significantly increased the soil nitrogen content and was positively correlated with SPAD value, Pn, IR, and yield (*p* < 0.05). Firstly, *Gemmatimonas* is a self-photosynthesising genus that contains an abundance of bacterial chlorophyll and can reduce soil N_2_O to nitrate [49].

*Bradyrhizobium* can form a nitrogen-fixing symbiosis with rhizobacteria from soybean roots, increasing soil nitrogen content and impacting photosynthetic properties [50]. In contrast, *Saccharimonadales* exhibited a significant negative correlation (*p* < 0.05) with Pn, SPAD, CAT, and yield but a significant positive correlation with plant NR and NIR. This could be attributed to the fact that this genus is involved in denitrification and phosphorus removal, which leads to the loss of soil nitrogen. This reduces the nitrogen uptake capacity of soybeans, which then affects the photosynthetic characteristics [14]. The plant NR and NIR are rate-limiting steps in the reaction of nitrate reduction to ammonium. The reduction in nitrogen uptake capacity in soybeans promotes the rate of this reaction, which in turn affects their activities.

In this study, *Sphingomonas* had a highly significant negative correlation with soil pH but a significant positive correlation (*p* < 0.05) with LA, Pn, IR, and yield. There are two possible explanations for this. Firstly, the genus can remediate heavy metal pollution and play an important role in increasing soil fertility and promoting plant growth [38]. Secondly, the decomposition of soil carbohydrates (pentoses, hexoses, and disaccharides) by this genus can produce large amounts of acid during oxidation, which in turn can reduce soil pH [51]. Enrichment of *Candidatus_Solibacte* and *Candidatus_Udaeobacter* can decrease soil nitrogen content, and *Candidatus_Solibacte* can effectively decompose soil organic matter and reduce soil nitrate and nitrite by consuming soil carbon sources [52]. However, increased soil carbon content not only leads to reduced soil nitrogen mineralisation, enhanced nitrogen sequestration, and reduced nitrification but is also related to the fertility level of the soil itself.

Nitrogen mineralisation is strongly influenced by the C/N ratio of organic matter. The high C/N ratios of organic matter reduce the soil mineralisation rate, and the mineralised nitrogen is easily sequestered [53]. *Candidatus_Udaeobacter* can release antibiotics, which can induce other microorganisms to lyse and release nutrients [54]. This study found that the genus was significantly positively correlated with soil CAT, which could increase soil nitrogen levels to some extent. At the same time, it exhibited a highly significant negative correlation with soil UE, proving that enrichment of the genus could inhibit the ability of soil urease to hydrolyse urea, resulting in a decrease in soil nitrogen [55].

## 5. Conclusions

The findings revealed that under the maize–soybean intercropping pattern, the N application rate was 80 kg·ha^−1^ more favourable in bridging the yield gap with monocropped soybean and also had the highest photosynthetic characteristics and soil nitrogen content. The soybean yield was positively correlated with photosynthetic capacity, soil nitrogen, nitrogen-assimilating enzyme activity, and dominant soil microbial genera. The main pathways and modes of intercropping effects on soybean yield, as well as the gradient changes generated by different levels of N application, were comprehensively elucidated. Summarily, although soybean sacrifices part of its yield in the intercropping mode, yield can be improved by boosting the soil microenvironments and above-ground synergistic capacity via appropriate nitrogen application levels.

## Figures and Tables

**Figure 1 microorganisms-12-01220-f001:**
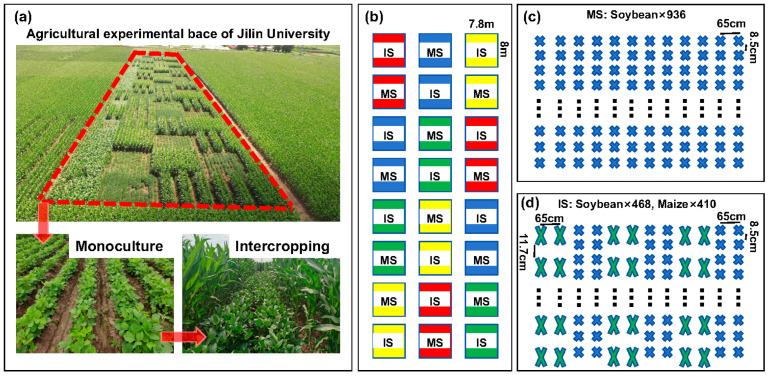
Maize–soybean intercropping experimental design. (**a**) aerial view; (**b**) planting distribution map, where red is the N0 level, blue is the N1 level, green is the N2 level, and yellow is the N3 level; (**c**) distribution of soybean monoculture (MS) trial plots; (**d**) distribution of maize–soybean intercropping trial plots. where 936, 468, and 410 are the number of crop plants in the plot; 65 cm refers to the spacing between rows, while 8.5 cm or 11.7 cm is the spacing between plants within a given row.

**Figure 2 microorganisms-12-01220-f002:**
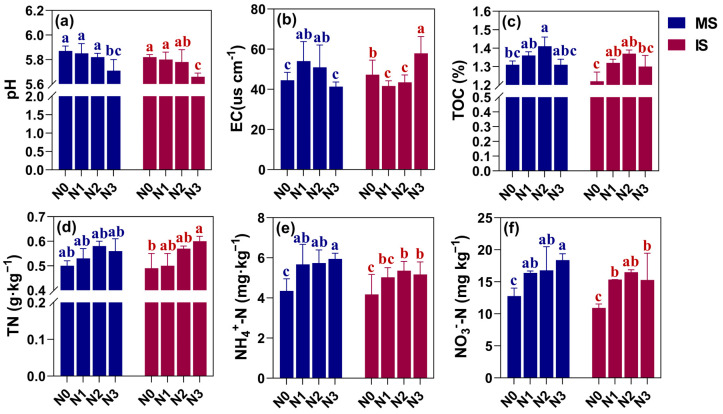
Effects of different cropping systems and N application rates on soil pH (**a**), EC (**b**), total organic carbon (TOC) (**c**), total nitrogen (TN) (**d**), ammonium nitrogen (NH_4_^+^-N) (**e**), and nitrate nitrogen (NO_3_^−^-N) (**f**). The bars represent the mean ± SE, n = 9 replicates. Small letters (a, b, c) on the error line indicate significant differences between treatments (*p* < 0.05). MS indicates monoculture soybean, and IS indicates intercropping soybean. N0–N3 indicate 0, 40, 80, and 120 kg N ha^−1^.

**Figure 3 microorganisms-12-01220-f003:**
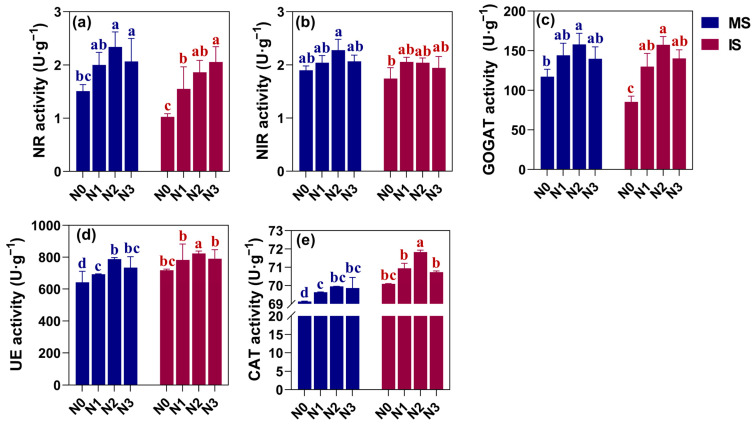
Effects of different cropping patterns and nitrogen application rates on nitrogen-assimilating enzyme activities. The bars represent the mean ± SE, n = 9 replicates. Small letters (a, b, c, d) on the error line indicate significant differences between treatments (*p* < 0.05). NR: plant nitrate reductase (**a**), NIR: plant nitrite reductase (**b**), GOGAT: plant glutamate synthase (**c**), UE: soil urease (**d**), and CAT: soil catalase (**e**). MS indicates monoculture soybean, and IS indicates intercropping soybean. N0–N3 indicate 0, 40, 80, and 120 kg N ha^−1^.

**Figure 4 microorganisms-12-01220-f004:**
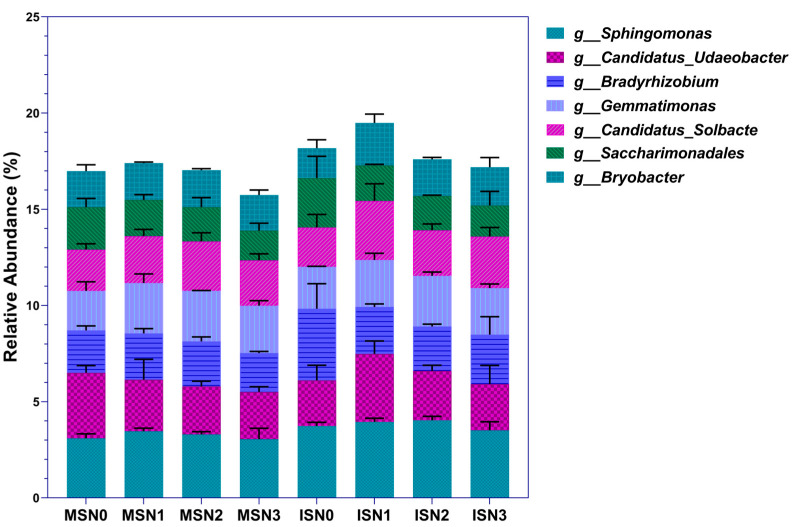
Horizontal community composition of bacterial genera under different cropping patterns and varying nitrogen application rates. The bars represent the mean ± SE, n = 9 replicates. MS indicates monoculture soybean, and IS indicates intercropping soybean. N0–N3 indicate 0, 40, 80, and 120 kg N ha^−1^.

**Figure 5 microorganisms-12-01220-f005:**
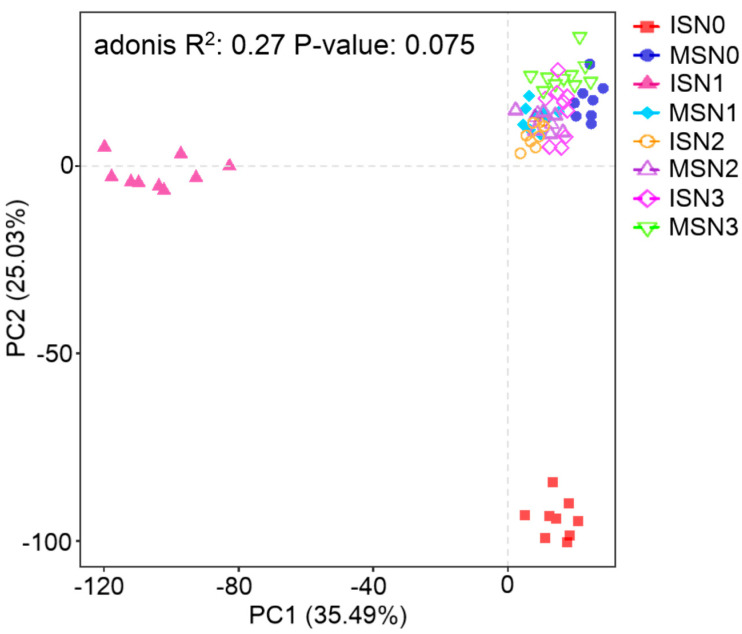
PCA of the bacterial community structure. n = 9 replicates. MS indicates monoculture soybean, and IS indicates intercropping soybean. N0–N3 indicate 0, 40, 80, and 120 kg N ha^−1^.

**Figure 6 microorganisms-12-01220-f006:**
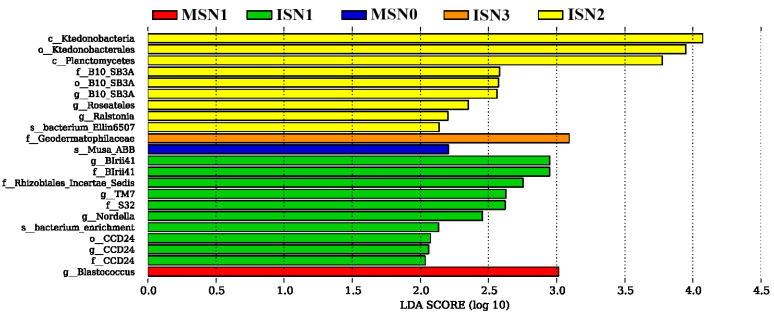
Determination of the main differing species in bacterial communities under different cropping patterns and nitrogen application rates using linear discriminant analysis (LDA) effect sizes (LEfSe). An LDA score ≥ 2 indicates a different species, i.e., a statistically different biomarker; the length of the bar represents the effect size of the significantly different species. MS indicates monoculture soybean, and IS indicates intercropping soybean. N0–N3 indicate 0, 40, 80, and 120 kg N ha^−1^.

**Figure 7 microorganisms-12-01220-f007:**
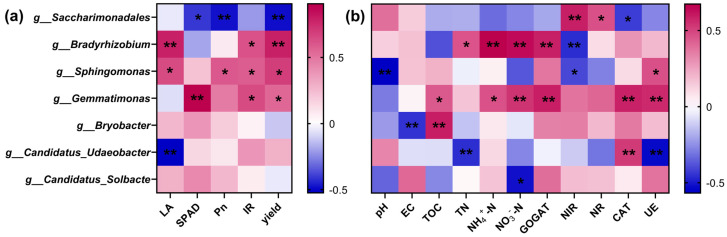
Correlation analysis of soil dominant bacterial genera with photosynthetic characteristics and yield (**a**) and soil factors (**b**), red indicates positive correlation, blue indicates negative correlation, and asterisks indicate significant correlation, * *p* < 0.05, ** *p* < 0.01. LA, Pn, and IR denote soybean leaf area, photosynthesis rate, and transpiration rate, respectively; TOC, TN, NH_4_^+^-N, and NO_3_^−^-N denote soil total organic carbon, total nitrogen, ammonium nitrogen and nitrate nitrogen, respectively; NR, NIR and GOGAT denote plant nitrate reductase, nitrite reductase and glutamate synthase, respectively; UE and CAT denote soil urease and catalase, respectively.

**Figure 8 microorganisms-12-01220-f008:**
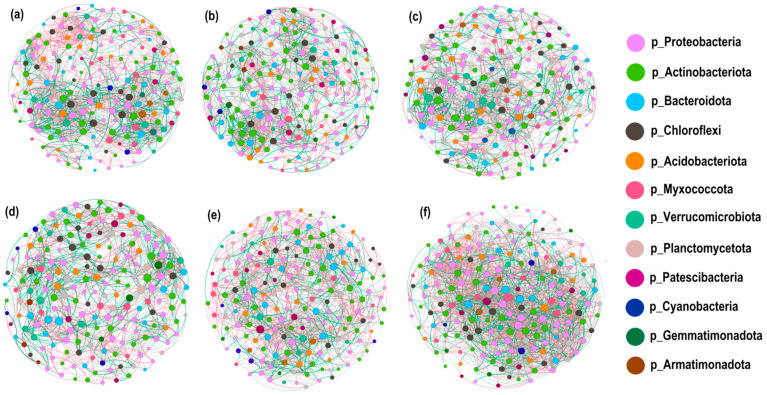
Co-occurrence network model of soil bacterial communities under different nitrogen application rates and cropping patterns. (**a**–**d**) represent different nitrogen application levels (N0, N1, N2, and N3) (MS and IS fitted by 97% similarity); (**e**) Soybean mono-cropping and (**f**) intercropping (N0, N1, N2, and N3 fitted by 97% similarity) for different cropping modes (N0–N3 fitted by 97% similarity) (**e**,**f**). The circles indicate different species (genus level), the size of the circle reflects the average abundance of the species, the line segments between the circles indicate a correlation between two species, and the thickness of the line segments indicates the degree of correlation between the two species. Red lines denote a positive correlation, and green lines denote a negative correlation. N0–N3 indicates 0, 40, 80, and 120 kg N ha^−1^.

**Figure 9 microorganisms-12-01220-f009:**
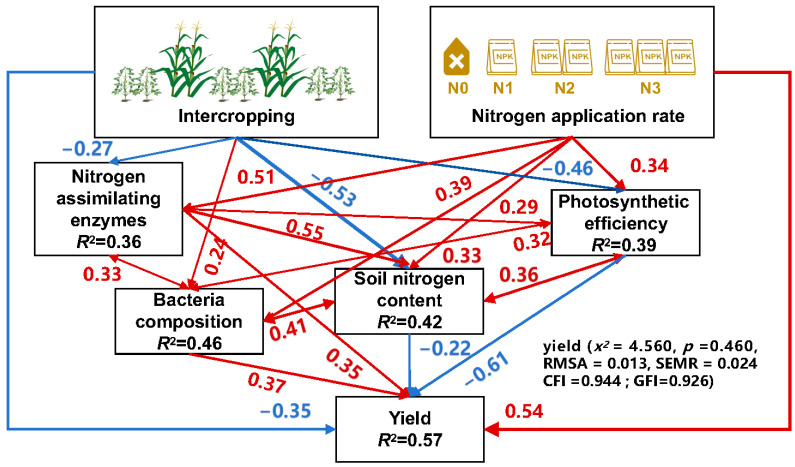
Structural equation modelling (SEM) illustrating the effects of intercropping and N application rates on photosynthetic characteristics, soil N, N-assimilating enzymatic activities, bacterial community composition, and yield in soybean. CFI and GFI denote the comparative fit index, and RMSEA denotes the root mean square error of approximation. N0–N3 indicate 0, 40, 80, and 120 kg N ha^−1^.

**Table 1 microorganisms-12-01220-t001:** Statistics of sequencing instruments and reagents.

Types	Instruments/Reagents	Producers	Specification/Model/Lot Number
Amplicon extraction	MoBio PowerSoil DNA Isolation Kit (100)	QIAGEN	100 times
Amplifier amplification	KAPA 2G Robust Hot Start Ready Mix	KAPA	
	ABI 9700 PCR	ABI	
Amplicon purification	Agencourt^®^ AMPure^®^ XP	Beckman Coulter	Dispense 45 mL/bottle, total 450 mL/bottle
Amplicon building	NEBNext Ultra II DNA Library Prep Kit	NEB	96 reactions
	Agencourt^®^ AMPure^®^ XP	Beckman Coulter	Dispense 45 mL/bottle, total 450 mL/bottle
	ABI 9700 PCR	ABI	
Library quality control instruments	Bioanalyzer (Agilent 2100)	Agilent	DE13806339
	Biomolecule Analyzer (Labchip GX)	PerkinElmer	
	ABI Qpcr	ABI	Step One Plus
Library quality control reagents	Agilent DNA 1000 Kit	Agilent	300 samples
	HT DNA-Extended Range LabChip	PerkinElmer	
	KAPA Library Quantification Kit	KAPA	500 times
Sequencing equipment	High-throughput second-generation sequencer	illumina	MiSeq
Sequencing reagents	MiSeq^®^ Reagent Kit v3 (600 cycle) (PE300)	illumina	
	MiSeq Reagent Kit v2 (500 cycle)	illumina	

**Table 2 microorganisms-12-01220-t002:** Effects of different cropping patterns (C) and nitrogen application rates (N) on photosynthetic characteristics and soybean yield.

Treatment		Pn (umol·m^2^·s^−1^)	IR (mmol·m^2^·s^−1^)	SPAD *	LA (dm^2^)	Yield (kg·ha^−1^)
MS *	N0	22.17 ± 3.68 b	10.98 ± 1.05 b	45.0 ± 3.7 cd	10.07 ± 0.51 b	1975 ± 121 c
	N1	26.33 ± 2.30 a	12.61 ± 1.54 a	49.1 ± 1.5 a	11.82 ± 0.52 ab	2268 ± 90 bc
	N2	24.69 ± 0.88 ab	12.55 ± 1.49 a	47.1 ± 2.2 b	14.12 ± 1.50 ab	2497 ± 123 b
	N3	25.48 ± 3.71 ab	12.08 ± 1.71 a	46.0 ± 3.1 bc	11.46 ± 3.12 ab	2562 ± 117 a
IS *	N0	20.55 ± 2.32 b	10.24 ± 2.13 b	44.4 ± 2.1 d	15.26 ± 3.76 a	1718 ± 79 d
	N1	21.88 ± 2.97 b	10.69 ± 1.28 b	45.4 ± 2.7 cd	16.22 ± 3.47 a	1931 ± 127 c
	N2	21.82 ± 3.16 b	10.84 ± 1.68 b	48.7 ± 3.6 a	16.34 ± 1.44 a	2256 ± 81 bc
	N3	23.94 ± 4.09 ab	10.56 ± 1.94 b	47.1 ± 3.3 b	15.97 ± 1.85 a	2241 ± 44 bc
Results of the two-way ANOVA test (F)						
	N	2.563 ns	1.085 ns	12.794 **	1.567 ns	12.934 **
	C	8.564 **	10.327 **	1.020 ns	24.616 **	15.273 **
	N * C	0.598 ns	0.290 ns	9.192 **	0.948 ns	9.928 **

Pn indicates net photosynthesis rate, IR indicates transpiration rate, and LA indicates leaf area. MS indicates monoculture soybean, and IS indicates intercropping soybean. N0–N3 indicate 0, 40, 80, and 120 kg N ha^−1^. The data indicate the mean ± SE, n = 9 replicates. The a, b, c, and d lettering indicate differences between treatments (*p* < 0.05). ns indicates no significant differences. An asterisk indicates that the effect factor had a significant effect on the outcome variable, * *p* < 0.05, ** *p* < 0.01.

**Table 3 microorganisms-12-01220-t003:** Alpha diversity statistics of bacterial communities under different cropping patterns (C) and nitrogen application rates (N).

Treatment		Chao1 Index	Shannon Index	PD_Whole_Tree	Goods_Coverage
MS	N0	8260 ± 764 ab	10.43 ± 0.06 bc	473.8 ± 6.09 a	0.97 a
	N1	8130 ± 292 b	10.41 ± 0.05 bc	463.9 ± 8.38 b	0.96 a
	N2	7908 ± 160 c	10.44 ± 0.04 bc	455.7 ± 5.41 b	0.97 a
	N3	7847 ± 254 c	10.38 ± 0.04 c	442.6 ± 9.32 c	0.97 a
IS	N0	8271 ± 589 ab	10.64 ± 0.00 a	475 ± 7.70 a	0.97 a
	N1	8312 ± 280 a	10.58 ± 0.06 ab	473.4 ± 2.13 a	0.96 a
	N2	8418 ± 322 a	10.51 ± 0.04 b	477.6 ± 0.24 a	0.97 a
	N3	7977 ± 134 c	10.34 ± 0.01 c	446.1 ± 12.42 c	0.97 a
Results of the two-way ANOVA test (F)					
	N	9.732 **	11.283 **	9.965 **	0.928 ns
	C	14.821 **	10.487 **	5.728 *	1.023 ns
	N * C	8.382 **	10.002 **	1.829 ns	0.892 ns

MS indicates monoculture soybean, and IS indicates intercropping soybean. N0–N3 indicate 0, 40, 80, and 120 kg N ha^−1^. The data indicate the mean ± SE, n = 9 replicates. The a, b, and c lettering indicate differences between treatments (*p* < 0.05). ns indicates no significant differences. An asterisk indicates that the effect factor had a significant effect on the outcome variable, * *p* < 0.05, ** *p* < 0.01.

**Table 4 microorganisms-12-01220-t004:** Indices of topological properties of the soil bacterial community co-occurrence network.

Treatment	Total Nodes	Edge	Positive (%)	Negative (%)	Average Degree	Average Weight	Cluster Coefficient	Modularity
N0 (MS, IS) *	200	1721	54.21	45.79	17.21	15.15	0.50	0.52
N1 (MS, IS)	200	1407	50.18	49.82	14.07	12.36	0.50	0.54
N2 (MS, IS)	200	1515	50.69	49.31	15.15	13.27	0.49	0.53
N3 (MS, IS)	200	1381	51.41	48.59	13.81	12.14	0.49	0.56
MS (N0–N3)	200	1342	54.92	45.08	13.42	13.28	0.34	0.45
IS (N0–N3)	200	2314	52.64	47.26	23.14	16.16	0.43	0.36

* MS indicates monoculture soybean, and IS indicates intercropping soybean. N0–N3 indicate 0, 40, 80, and 120 kg N ha^−1^.

## Data Availability

The datasets generated and analysed during the current study are available from the corresponding author upon reasonable request.

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
