# Peer review of "Macrogenomics-Based Analysis of the Effects of Intercropped Soybean Photosynthetic Characteristics and Nitrogen-Assimilating Enzyme Activities on Yield at Different Nitrogen Levels"

_microorganisms, 2024, doi:10.3390/microorganisms12061220_

Round 1

Reviewer 1 Report

Comments and Suggestions for Authors

The manuscript, entitled: "Macrogenomics-based analysis of the effects of intercropped soybean photosynthetic characteristics and nitrogen-assimilating enzyme activities on yield at different nitrogen levels". It presents the results of a well-constructed intercropping experiment using corn and soybean plants. It intensively investigates soil chemical and microbiological changes. It makes important findings regarding the development of the microflora and the method of cultivation, as well as nitrogen replacement levels.

The introduction and the material and method chapters have been developed in sufficient detail. The presentation of the results is sufficiently detailed. It analyzes well the changes that occur during intercropping cultivation, as well as the microorganism groups that can be linked to nitrogen replacement levels. It makes correct statements about the role of individual bacterial and fungal genera in determining the circulation and mobilization of nitrogen.

Improving the level and sustainability of soybean production can be helped by precisely determining the effects that occur in plant physiology and yield changes.

After correcting the suggested typographical errors, I recommend publishing the manuscript in the form of a scientific article.

Comments on the Quality of English Language

The English language of the manuscript is almost error-free. It contains minor typos and letter omissions. I recommend improving them, e.g. in row 30 and 694 the name Solbacte is written instead of Solibacter.
In line 429, I recommend correcting nitrogen water to nitrogen level.

Reviewer 2 Report

Comments and Suggestions for Authors

the manuscript entitled <Macrogenomics-based analysis of the effects of intercropped 2 soybean photosynthetic characteristics and nitrogen-assimilat- 3 ing enzyme activities on yield at different nitrogen levels>

presents important results on soybean cultivation showing that  the maize-soybean intercropping pattern, the N level (80 kg-ha-1) was better  in bridging the yield gap with monocropped soybean, and also had the highest photosynthetic characteristics and soil nitrogen content. 710 The soybean yield was correlated with photosynthetic capacity, soil nitrogen, nitrogen- 711 assimilating enzyme activity, and dominant soil microbial genera. The main pathways  and modes of intercropping effects on soybean yield, as well as the gradient changes generated by different levels of N application, were discused,  although soybean sacrifices part of its yield in intercropping mode, yield can be improved by boosting the soil microenvironments andaboveground synergistic capacity through appropriate nitrogen application levels. 

The topic is extremely important, as soybean is one of the main crops worldwide.

the paper is well analyzed and presented, including well presented figures, however, minor details will improve the manuscript:

To use soil N and not:N2 as N2 is gaseous )line708)

line711: + correlated? please, indicate this.

line301: ckheck : TOC, FIG.2 LEGEND

LINE131:THE REmained N fertilizer: It can be better explained.
